# Effect of Polycondensation Catalyst on Fiber Structure Development in High-Speed Melt Spinning of Poly (Ethylene Terephthalate)

**DOI:** 10.3390/polym11121931

**Published:** 2019-11-22

**Authors:** Eun Seon Kim, Hyun Ju Oh, Hyun-Joong Kim, Chun Gi Kim, Seong Yoon Park, Young Gyu Jeong, Wan-Gyu Hahm

**Affiliations:** 1Department of Chemical Engineering, Hanyang University, Seoul 04763, Korea; kes1207@gmail.com; 2Technical Textile R&D Group, Korea Institute of Industrial Technology, Ansan 15588, Korea; hjoh33@kitech.re.kr; 3Business Department, Samwon Ind. Co. Ltd., Ansan 15612, Korea; hjkim@samwon21.com; 4R&D Center, Huvis Corporation, Daejeon 34326, Korea; polyhoon@huvis.com (C.G.K.); psy@huvis.com (S.Y.P.); 5Department of Advanced Organic Materials and Textile System Engineering, Chungnam National University, Daejeon 34134, Korea; ygjeong@cnu.ac.kr

**Keywords:** poly (ethylene terephthalate), catalyst, titanium, crystallization, melt spinning

## Abstract

We conducted a preliminary study on fiber structural development in the high-speed melt spinning of environmentally friendly polyethylene terephthalate (Ti-PET) synthesized with 25 ppm of titanium-based catalyst, which was compared with conventional PET (Sb-PET) synthesized with 260 ppm of antimony-based catalyst. Gel permeation chromatography of Ti- and Sb-PET resins of intrinsic viscosity 0.63 confirmed that both resins have similar molecular weights and distributions. However, differential scanning calorimetry revealed that the Ti-PET resin exhibited a lower melt–crystallization peak and isothermal melt-crystallization rate than the Sb-PET resin. High-speed melt spinning of the Ti- and Sb-PET was possible up to a spinning velocity of 6 km/min. Two-dimensional wide-angle X-ray diffraction analyses showed that the molecular orientation of the obtained as-spun Ti- and Sb-PET fibers increased with spinning velocity, and a highly oriented, crystalline structure by orientation-induced crystallization started to appear from 5 km/min. Notably, Ti-PET fibers showed a lower degree of crystalline structural development and lower tensile strength compared with Sb-PET fibers under the high-speed spinning conditions. Our results suggest that the catalyst in PET resins can act as nucleating agents in thermal- and orientation-induced crystallization, and that differences in catalyst content can influence PET fiber structure development under extreme conditions in high-speed melt spinning.

## 1. Introduction

Poly (ethylene terephthalate) (PET) is a thermoplastic polymer widely used in synthetic fibers and for various applications, such as films, bottles, and molded plastic parts [1,2,3]. Commercial PET resins are mostly produced via two-step polymerization, comprised of esterification and condensation using raw materials, such as ethylene glycol (EG) and terephthalic acid (TPA), or EG and dimethyl terephthalate (DMT). Antimony (Sb)-based compounds, such as antimony trioxide (Sb_2_O_3_), antimony acetate (Sb(CH_3_COO)_3_), and antimony glycolate (Sb_2_(OC_2_H_4_O)), have mainly been used as a polycondensation catalyst in the last few decades [4,5].

However, it has been reported that the Sb element, which is a heavy metal, is suspected to be a harmful substance that can cause diseases such as cancer in the human body, and recently, environmental regulations have been strengthened to prevent the use of Sb in clothes, food packaging, and beverages bottles [6]. Therefore, many manufacturers have focused on Sb-free catalysts using non-toxic metals, such as titanium, aluminum, magnesium, germanium, and phosphorus [7,8,9,10,11,12,13].

Titanium (Ti)-based catalysts are one of the most commercially available catalysts employed as Sb-based catalyst substitutes, and have attracted manufacturers’ attention due to their excellent performance resulting from their large surface area (due to their high porosities and particle sizes), environmental friendliness, and low cost compared to the other Sb-free catalysts [5,14,15]. Furthermore, Ti-based catalysts show good performance with relatively low catalyst loadings (TiO_2_ 10–30 ppm) compared to those of Sb-based catalysts (Sb_2_O_3_ 150–300 ppm) in polymerization [16,17,18], and form fewer by-products, such as oligomers, during PET melt spinning than Sb-based catalysts, [5,19,20,21]. Some researchers have reported that the low loading of Ti-based catalysts can be expected to improve the processability and product quality in spinning and drawing processes, due to decreased amounts of impurities (catalyst) in the polymer, and shift the melt–crystallization peak (*T*_mc_) of PET towards a lower temperature on the differential scanning calorimetry (DSC) cooling curve, because the catalyst can act as a nucleation agent [22].

The high-speed spinning process is one of the commercial melt spinning processes used to produce PET fibers. It is well known that main fiber structure development in the process manifests with orientation-induced crystallization by high-speed spinning, and obtained as-spun fibers have a highly oriented, large-size, crystalline lamellar structure [22,23,24]. Specifically, it has been reported that the fiber structure evolution is strongly affected by various factors, including spinning temperature, spinning draft, molecular weight of resin, and additives [25]. This suggests that high-speed melt spinning can be useful for systematically studying the characteristic structural development and crystallization behavior of polymer resins. However, there are few reports of study on the novel Ti-PET resins synthesized with a low content of Ti-based catalyst in the high-speed spinning process.

In this study, prior to various studies of novel Ti-PET resins synthesized with Ti-based catalyst, we conducted a preliminary investigation on the characteristic structural development of Ti-PET synthesized with a relatively low content (25 ppm) of Ti-based catalyst in comparison to a conventional PET (Sb-PET) synthesized with Sb-based catalyst (260 ppm) in high-speed melt-spinning. The molecular weight and thermal properties of the prepared Sb-PET and Ti-PET resins were characterized by gel permeation chromatography (GPC), thermogravimetric analysis (TGA), and DSC. The various as-spun PET fibers were analyzed by two-dimensional wide-angle X-ray diffraction (2D-WAXD), birefringence measurement, and mechanical measurements.

## 2. Materials and Methods

### 2.1. Materials

In this study, a Ti-based catalyst purchased from Sachtleben Chemie GmbH (Duisburg, Germany) was used as an alternative to Sb_2_O_3_. Novel Ti-PET and conventional Sb-PET resins with the same intrinsic viscosity (IV) of 0.63 were synthesized with 25 ppm of the Ti-based catalyst and 260 ppm of antimony trioxide (Sb_2_O_3_) catalyst, respectively, by the same polymerization process. The two-part polymerization process of direct esterification and polycondensation was performed using terephthalic acid (TPA) and ethylene glycol (EG) monomers, which were prepared and supplied by Huvis Co. Ltd. (Daejeon, Korea). Other additives, such as thermal stabilizer, were loaded on both polymers, and their contents were controlled to be the same during the polymerization.

### 2.2. High-Speed Melt Spinning

The PET resins, after drying over 12 h at 150 °C, were melted in an extruder with a single-screw diameter of 30 mm and extruded through a spinneret with a single hole of 0.5 mm diameter by a gear pump. The throughput rate of the polymer and spinning temperature were controlled at 5.0 g/min and 285 °C, respectively. High-speed melt spinning of Ti-PET and Sb-PET was conducted at spinning velocities from 2 to 6 km/min. The obtained as-spun fiber samples were named according to the catalyst type (Sb or Ti) and spinning velocity (km/min), as shown in Table 1.

### 2.3. Characterization

The intrinsic viscosity (IV) of the Ti- and Sb-PET resins and their obtained as-spun fibers were measured by an Ubbelohde viscometer (CT72, SI Analytics, Mainz, Germany). The samples were dissolved in a solvent composed of phenol and 1,1,2,2-tetrachloroethane (6:4 *w*/*w*) at 25 °C. The molecular distribution of polymerized PET resins prepared with Ti- and Sb-based catalysts was determined by GPC on a Tosoh EcoSEC HLC-8320 GPC (TOSOH Bioscience, Griesheim, Germany) system at 40 °C using an internal refractive index (RI) detector and two TSKgel columns (SuperAWM-H, TOSOH Bioscience, Griesheim, Germany). Hexafluoro-isopropanol (HFIP) was used as the solvent at a flow rate of 0.3 mL/min. The concentration of residual Sb element in the obtained as-spun PET fiber was analyzed by ICP-OES (inductively coupled plasma optical emission spectroscopy) (ULTIMA2, Jobin-Yvon Horiba, Tokyo, Japan) equipped with a photomultiplier tube (PMT) detector in an Ar gas atmosphere. The cooling and sheath gas rate conditions were 12 L/min and 0.1 L/min, respectively. The thermal properties were analyzed by TGA (Q500, TA Instruments, New Castle, DE, USA) in N_2_ gas at the rate of 10 °C/min. These results are summarized in Table 2.

The crystallization behavior of polymerized PET resins was characterized by DSC (DSC1, Mettler Toledo, Zurich, Switzerland) in non-isothermal and isothermal modes. For non-isothermal scanning, the samples were heated from 30 °C to 280 °C and cooled at a rate of 10 °C/min under a N_2_ atmosphere. For isothermal crystallization measurements, the samples were heated to 280 °C at a rate of 20 °C/min under a N_2_ atmosphere, maintained at 280 °C for 10 min, then rapidly cooled to a predetermined temperature (195 °C, 205 °C) at a cooling rate of 100 °C/min, and maintained at the predetermined temperature for 30 min, to monitor the crystallization behavior.

The structural evolution of the obtained as-spun PET fibers was analyzed by 2D-WAXD (Rigaku Denki Co., Tokyo, Japan) with a nickel-filtered CuK α radiation (λ = 0.154 nm) operated at 40 kV and 60 mA. 2D-WAXD patterns were accumulated through an image plate detector at a camera length of 51.5 mm. To measure the birefringence (∆*n*) of the obtained as-spun fibers, the diameter (D) and optical retardation (Γ_c_) of the fibers were measured by a polarization microscope (Axioplan, Carl-Zeiss, Oberkochen, Germany) equipped with a Berek compensator. The birefringence (∆*n*) was calculated with the following Equation (1):
(1)$$\Delta n=\frac{1}{D}(k\lambda +\Gamma_{c})$$
where λ is the wavelength (546 nm) of monochromatic light, and *k* is the order of the interference [26]. The mechanical properties of the obtained as-spun fibers were measured using an automatic single-fiber test system (Textechno, Favimat, Mönchengladbach, Germany) based on the ASTM D2256 standard testing method. The gauge length and crosshead speed were 20 mm and 20 mm/min, respectively. The samples were prepared in the form of single filaments. At least 20 specimens were evaluated, and average values obtained from each sample are reported.

## 3. Results and Discussion

### 3.1. PET Resins

To verify the elemental Sb content of polymerized PET resins, ICP-OES analysis was first conducted. As expected, approximately 244 ppm of residual elemental Sb was detected in the Sb-PET resin, as shown in Table 2, while it was not detected (ND) in the Ti-PET resin. IV, GPC, and TGA characterization results for prepared PET resins are also summarized in Table 2. Ti-PET materials showed similar IV values to Sb-PET, and the molecular weight (*M*_w_ and *M*_n_), and molecular distribution (*M*_w_/*M*_n_) of Ti-PET determined by GPC also showed similar values to those of Sb-PET. These results indicate that the Ti-based catalyst shows good performance for PET polymerization, despite its loading usage being approximately 10 times lower than that of the conventional Sb_2_O_3_ catalyst. In the Ti-PET TGA curve, the 1% and 10% weight loss temperatures are similar to those of Sb-PET, which also suggests that the Ti-PET material has thermal stability comparable to Sb-PET.

Figure 1 shows non-isothermal DSC thermogram curves for Ti-PET and Sb-PET. The melting temperature peaks (*T*_m_) on the first heating scan curves of the Ti- and Sb-PET resins appeared at approximately 255 °C. The reason that the cold crystallization peak (*T*_cc_) did not appear on the first heating curve of each PET seems to be because the PET resins had some crystalline structure because of insufficient cooling during the pelletizing process after the polymerization. However, it is interesting to note that the melt–crystallization peak (*T*_mc_) on the first cooling scan curve of Ti-PET was at 162.8 °C, or approximately 16 °C lower than that of Sb-PET. Grebe R. et al. [14] reported that catalyst particles can act as a nucleating agent during crystallization of PET, and thus PET synthesized with Ti-based catalysts at approximately 20 ppm showed a relatively lower *T*_c_ peak on the DSC cooling curve than PET synthesized with Sb_2_O_3_ catalyst at approximately 200 ppm. These results seem to be consistent with our non-isothermal DSC results.

To verify the crystallization behaviors of Ti-PET versus Sb-PET in detail, isothermal crystallization experiments were also performed and characterized by DSC measurements. Figure 2a shows the obtained time-dependent isothermal crystallization curves of the Ti- and Sb-PET resins. It was found that the exothermic crystallization peaks of Sb-PET at each isothermal temperature appeared much earlier than the corresponding Ti-PET peaks, which supports that the overall isothermal melt–crystallization rate of Sb-PET is much higher than that of the Ti-PET.

The time-dependent relative crystallinity fraction (*X_t_*) of PET resins with residual catalyst can be evaluated by the integration of the area under the exothermic peak from the onset point to the end point, as shown in Figure 2b. The crystallization half time (*t*_1/2_) obtained from the relative crystallization fraction graph of Sb-PET was also significantly shorter, approximately 1.7–2.0 times shorter than that of Ti-PET (Table 3).

The isothermal crystallization behavior of PET resins with a residual catalyst was analyzed using the Avrami Equation:1 − *X*(*_t_*) = exp(−*Kt^n^*),(2)
where *X*(*_t_*) is the relative crystallinity fraction at time *t*, *K* is the temperature-dependent crystallization rate constant, and *n* is the Avrami exponent. Both *K* and *n* are related to nucleation and crystal growth [15]. Equation (2) is rewritten in the double logarithmic form as follows:log[−ln(1 − *X*(*t*))] = *n*log(*t*) + log(*K*),(3)
where *K* and *n* were determined from the slopes and intercepts of the best linear fit for the plots (log[−ln(1 − *X*(*t*))] versus log(*t*) of Figure 2c. The results are summarized in Table 3. In this work, there was an inflection point in each plot for Sb- and Ti-PET, indicating that the isothermal crystallization of PET occurs in two steps, thus, we segmented the linear fit, and calculated the *n* and *K* values either side of the inflection point.

The *n*_1_ of the primary crystallization showed lower values than the *n*_2_ of the secondary crystallization for all plots for both Sb- and Ti-PET. Furthermore, the *n*_1_ and *n*_2_ values for Ti-PET crystallized isothermally at 195 °C and 205 °C were consistent with the values for Sb-PET, suggesting that the isothermal crystallization mechanism for Ti-PET is similar to that for Sb-PET [27].

On the other hand, *K*_1_ and *K*_2_, related to the rate of crystallization, tended to increase with decreasing isothermal crystallization temperature. It is worth noting that the *K*_1_ and *K*_2_ values for Sb-PET were an order of magnitude higher than the values for Ti-PET.

Considering the non-isothermal and isothermal DSC results as a whole, it is shown that residual PET resin catalysts can, as a nucleating agent, influence the temperature-induced crystallization behavior of PET resins, and that accordingly, their crystallization rate can increase with catalyst content, although low.

### 3.2. PET Fiber in High-Speed Melt Spinning

In general, the cooling rate of fiber in melt spinning is very high, and in the case of polymer resins with low crystallization rates, such as PET, the extruded molten fiber can simply cool into a non-crystalline structure before crystallization occurs in the spinning line. However, it is well established that the crystallization rate of polymers is greatly enhanced by the presence of molecular orientation, and that high spinning tension of high-speed spinning can dramatically accelerate the crystallization rates of fiber [19]. The authors and many other researchers have also reported that orientation-induced crystallization is associated with neck-like fiber deformation in high-speed spinning, and the mechanism of structure development near the deformation has been discussed by systematically referring to various studies [19].

In this study, to verify the characteristic orientation-induced crystallization and structure development of Ti-PET fiber in the melt spinning process, high-speed melt spinning of Ti-PET was performed up to 6 km/min, and compared with Sb-PET. Notably, Ti-PET showed a similar spinnability to Sb-PET in high-speed melt spinning, and the obtained Ti-PET as-spun fiber showed a similar IV drop (difference in IV values before and after melt spinning) and IV values to the Sb-PET as-spun fibers (Table 2).

Figure 3 shows the change in birefringence (∆*n_B_*) for the Ti- and Sb-PET as-spun fibers as a function of the spinning velocity. The birefringence, indicative of the degree of molecular orientation of the as-spun PET fibers, increased with increasing spinning velocity, and showed a sigmoidal curve that increased rapidly at a spinning velocity between 4 and 5 km/min. This suggests that orientation-induced crystallization of fibers due to high-speed spinning started to occur between 4 and 5 km/min [19]. However, it was not possible to verify any difference between the structure developments of Sb- and Ti-PET as-spun fibers via the birefringence, because their respective birefringence values were within the error range at all spinning velocities.

The mechanical properties, such as tensile strength and elongation at break, of the Ti- and Sb-PET as-spun fibers obtained at spinning velocities of 4–6 km/min are summarized in Figure 4. As expected, the tensile strength of the as-spun fibers increased, while the elongation at break rapidly decreased as the spinning velocity increased (Figure 4b). The stress–stain curves in Figure 4a show the detailed characteristic relationships between fiber structural evolution and mechanical properties for several spinning velocities. The fibers spun at 4 km/min showed high elongation properties and some constant load elongation (i.e., ND, natural drawing) after the yield point, while from the spinning velocity of 5 km/min, there was no ND zone and only a strain-hardening curve was observed after the yield point, suggesting that the fiber structure was fully oriented. However, Ti-PET fibers spun at 5 and 6 km/min indicated slightly lower average tensile strength, indicating a lower degree of structural development than those of the Sb-PET as-spun fibers.

WAXD analysis was conducted to more accurately and systematically investigate the differences in structure development between as-spun high-speed melt spun Ti- and Sb-PET fibers. The 2D-WAXD patterns of the Ti- and Sb-PET as-spun fibers prepared at various spinning velocities are shown in Figure 5. An isotropic amorphous halo was observed at the spinning velocity of 2 km/min, and a distinct anisotropic distribution started to appear at the spinning velocity of 3 km/min, due to the molecular orientation resulting from the drawing direction of the fiber. The crystalline peaks of the Ti- and Sb-PET fibers in the 2D-WAXD patterns caused by orientation-induced crystallization both appeared at the spinning velocity of 5 km/min, and the sharpness of crystalline peak increased with spinning velocity [19]. It is interesting that the crystalline peaks in the equatorial direction of Ti-PET as-spun fibers obtained at high-speed spinning velocities of 5 and 6 km/min showed lower intensities than those of the corresponding Sb-PET fibers, as shown in Figure 6.

Various fiber structures from the 2D-WAXD patterns were investigated in detail as follows. Figure 7a shows the result of a typical peak fitting of the equatorial intensity profiles obtained from the 2D-WAXD pattern of Ti-PET-6 resolved into peaks corresponding to each phase and used to calculate the apparent crystallinity (*X_cr_*) of the crystalline as-spun fiber. A broad intensity profile diffracted by the non-crystalline phase at 2θ = 21.3° was first estimated based on the equatorial intensity profiles of the highly oriented non-crystalline as-spun fiber (Ti-PET-4 and Sb-PET-4), and then, the diffraction peaks corresponding to the (010), (−110), and (100) planes of the PET crystalline structure were determined. The apparent crystallinity (*X_cr_*) was calculated with Equation (4) [22]:
(4)$$X_{cr}=\frac{X_{total}-\left(X_{meso}+X_{am}\right)}{X_{total}}$$
where *X_total_* is the total area (*X_cr_* + *X_meso_* + *X_am_*) of the equatorial intensity profile, and (*X_meso_* + *X_am_*) is the area of the non-crystalline phase consisting of the oriented non-crystalline phase (mesophase, *X_meso_*) and non-oriented non-crystalline phase (amorphous phase, *X_am_*).

The crystallite size (*L_hkl_*) for each *hkl* reflection plane was estimated with Scherrer’s equation, Equation (5):
(5)$$L_{hkl}=\frac{K_{s}\lambda }{\beta_{hkl}\cos\theta }$$
where *K_S_* is the correction factor (*K_S_* = 0.9), λ is the X-ray wavelength (λ = 0.154 nm), θ is the Bragg angle of the diffraction peak *hkl*, and β*_hkl_* is the intrinsic half width of the peak *hkl*. The interplanar spacing (*d*_hkl_) of each *hkl* reflection plane and the number of repeating units per crystal (*N*) were calculated with the Bragg equation (*n*λ = 2*d*_hkl_ sinθ) and *L_hkl_*/*d*_hkl_, respectively [21].

The crystal orientation factor (*f*_c_) of an as-spun fiber was estimated by the azimuthal intensity profiles of the 2D-WAXD reflection lines from the (010) plane and the quantities of these planes were obtained from Equation (6):
*f*_c_ = (180 − Φ)/180
sinΦ = cosθ × sin*X*_E_(6)
where Φ is the inclination of the c-axis to the fiber axis, θ is the Bragg angle of the (010) plane, and *X*_E_ is the half width of the azimuthal intensity profile for the (010) plane on the equator [23].

Characterization results are summarized in Table 4. In the case of crystalline Ti- and Sb-PET fibers spun at 5–6 km/min, the interplanar spacing (*d*_hkl_) of the (010), (100), and (−103) planes decreased, which indicates that the crystalline perfection in the fibers improved, and the crystallite size (*L*_hkl_), number of repeating units per crystal (*N*), and crystalline orientation factor (*f*_c_) increased as the spinning velocity increased. It is notable that the crystallite sizes of the Ti-PET fibers were relatively small compared to those of the Sb-PET fibers.

Figure 7b shows the change in the fractions of the crystalline (*X_cr_*), mesophase (*X_meso_*), and amorphous phase (*X_am_*) of Ti- and Sb-PET as-spun fibers as a function of spinning velocity. The fraction of the amorphous phase (*X_am_*) of the Ti- and Sb-PET as-spun fibers decreased linearly as the spinning velocity was increased, and the Ti-PET fibers exhibited similar values to the Sb-PET fibers in all spinning velocity ranges. However, the fraction of mesophase (*X_meso_*) increased with spinning velocity, but drastically decreased after the crystalline structure started to appear and increased from 5 km/min. These results indicate that the mesophase acts as a precursor between the amorphous phase and crystalline structure during orientation-induced crystallization in high-speed spinning [1,19,20,27]. Kikutani [19] also reported that structure development by orientation-induced crystallization, starting before the end of the neck-like deformation in high-speed spinning, occurs through the stages of molecular orientation, nucleation, and crystallization, as shown in Figure 8.

Considering the results of previous studies, it is worth noting that Ti-PET fibers showed a relatively low crystallinity (*X_cr_*) and high mesophase fraction (*X_meso_*) compared to Sb-PET fibers at the high-speed spinning velocities of 5 and 6 km/min, despite the similarity between their amorphous fractions (*X_am_*). In other words, these results suggest that the rate of crystal development via the mesophase in Ti-PET fibers by orientation-induced crystallization in high-speed spinning is relatively lower than that of Sb-PET fibers.

We therefore speculate that one of the reasons for the decreasing crystallization rate of Ti-PET fibers in high-speed spinning could be related to the low catalyst content, which can act as a nucleation agent during the orientation-induced crystallization.

In addition, if we assume that molten Ti-PET fiber with a lower catalyst content might solidify more slowly than Sb-PET fiber due to the deceleration of crystallization in high-speed spinning, these behaviors can result in a decrease of the maximum strain rate ((T-*D*_a_ − T-*D*_b_)/T-*L*_n_) in the vicinity of the neck-like deformation of the Ti-PET fiber, or an increase of the length of the deformation (T-*L*_n_), as shown in Figure 8. In particular, these results can result in a decrease in the total solidification length (T-*L*_s_), which can decrease the spinline tension (spinning stress) due to a decrease in air drag force for the corresponding Sb-PET fiber spinning velocity [25].

This assumption can explain why the crystalline Ti-PET fibers spun at 5 and 6 km/min showed a slightly lower mechanical strength (Figure 4) and lower molecular orientation of the mesophase (β_meso_ in Table 4), which can be estimated from the half width of the azimuthal intensity profile at 2θ ≈ 21.0° in the 2D-WAXD pattern, compared to Sb-PET fibers. However, further research is required to verify this assumption, and we will conduct further research, such as an analysis of fiber formation by high-resolution on-line diameter measurement in high-speed spinning of Ti-PET resins prepared with various Ti-based catalysts.

## 4. Conclusions

In this study, the fiber structure development of an environmentally friendly PET (Ti-PET) synthesized with 25 ppm of titanium-based catalyst was investigated in comparison to a conventional PET (Sb-PET) synthesized with 260 ppm of antimony-based catalyst. Both the molecular weight and distribution of synthesized Ti- and Sb-PET resins were similar. However, the Ti-PET resin showed a lower melt-crystalline temperature (*T*_mc_) and crystallization rate (*K*) than the Sb-PET in the DSC results. High-speed melt spinning of Ti-PET resin also showed the characteristic fiber structural development behavior observed in Sb-PET resin. 2D-WAXD results of as-spun fibers obtained at 5 and 6 km/min of high-speed spinning indicate that the crystallite size and crystalline perfection of Ti-PET as-spun fiber due to orientation-induced crystallization was relatively low compared to that of Sb-PET as-spun fibers. Furthermore, the tensile strength of the crystalline Ti-PET fibers showed a greater tendency to decrease than that of the Sb-PET fibers. From these results, we suggest that the catalyst in PET resins can act as nucleating agent in thermal- and orientation-induced crystallization, and the difference in the content of the added catalyst can influence PET fiber structure development in the extreme condition of high-speed melt spinning, even if the content is low. In addition, these results indicate that the resin conditions of the high-speed melt spinning process, which produces fibers by the orientation-induced crystallization method, should be more precisely controlled, because their fiber structure and thermo-mechanical properties can be more affected by small changes in resin compared to conventional low-speed melt spinning.

## Figures and Tables

**Figure 1 polymers-11-01931-f001:**
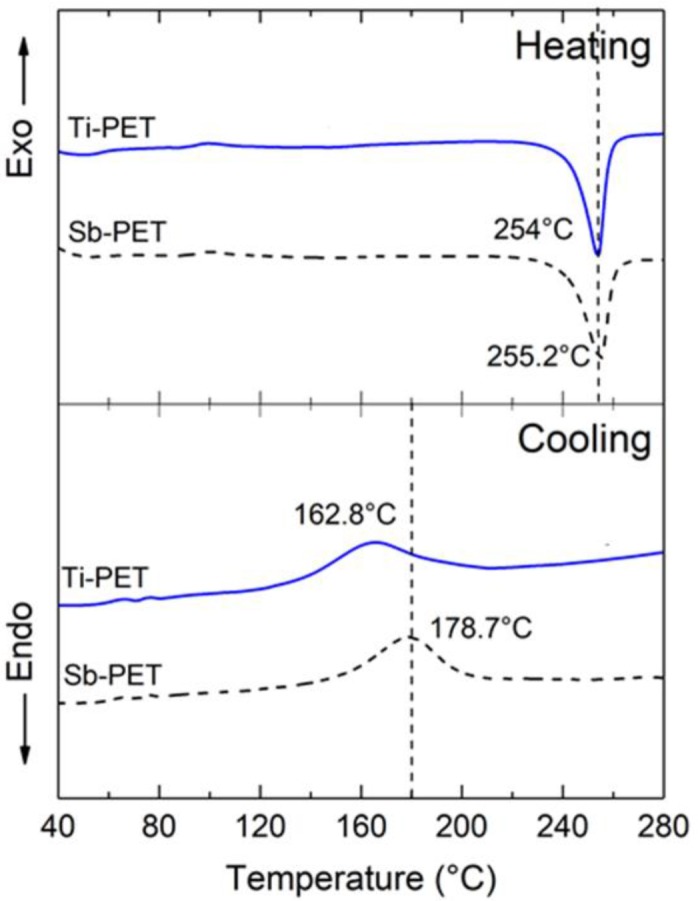
Non-isothermal differential scanning calorimetry (DSC) thermograms of the PET resins polymerized with Ti and Sb catalysts.

**Figure 2 polymers-11-01931-f002:**
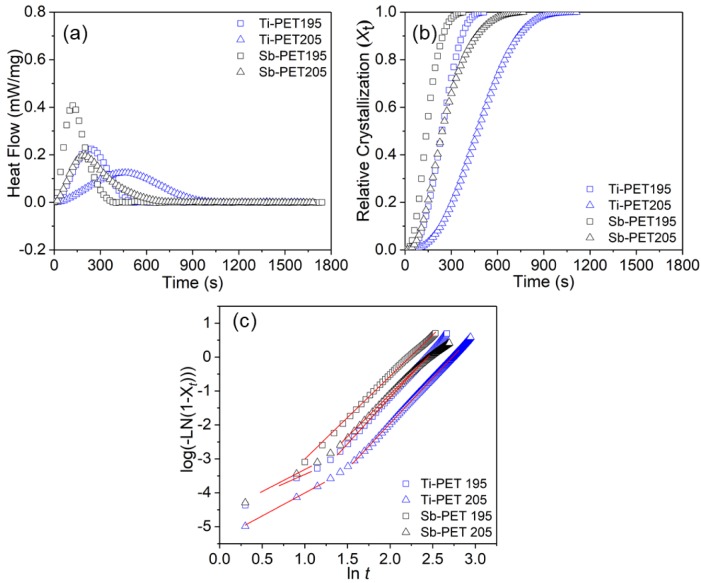
Characteristics of Ti-PET and Sb-PET resins at isothermal temperatures of 195 °C and 205 °C. (**a**) The total isothermal crystallization peak; (**b**) The variation in fractional crystallinity *(X_t_*); (**c**) The linear plot of log[−ln(1 − *X_t_*)] versus log(*t*).

**Figure 3 polymers-11-01931-f003:**
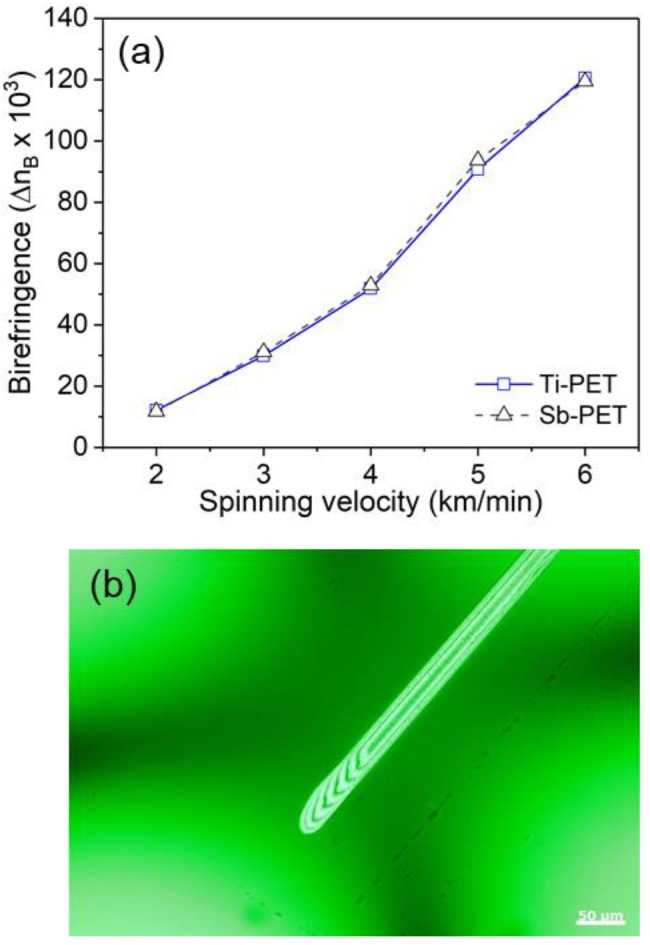
(**a**) Birefringence curves of the Ti- and Sb-PET as-spun fibers as a function of spinning velocity; (**b**) interference image of Ti-PET-6 fiber observed by polarization microscopy showing seven fringes.

**Figure 4 polymers-11-01931-f004:**
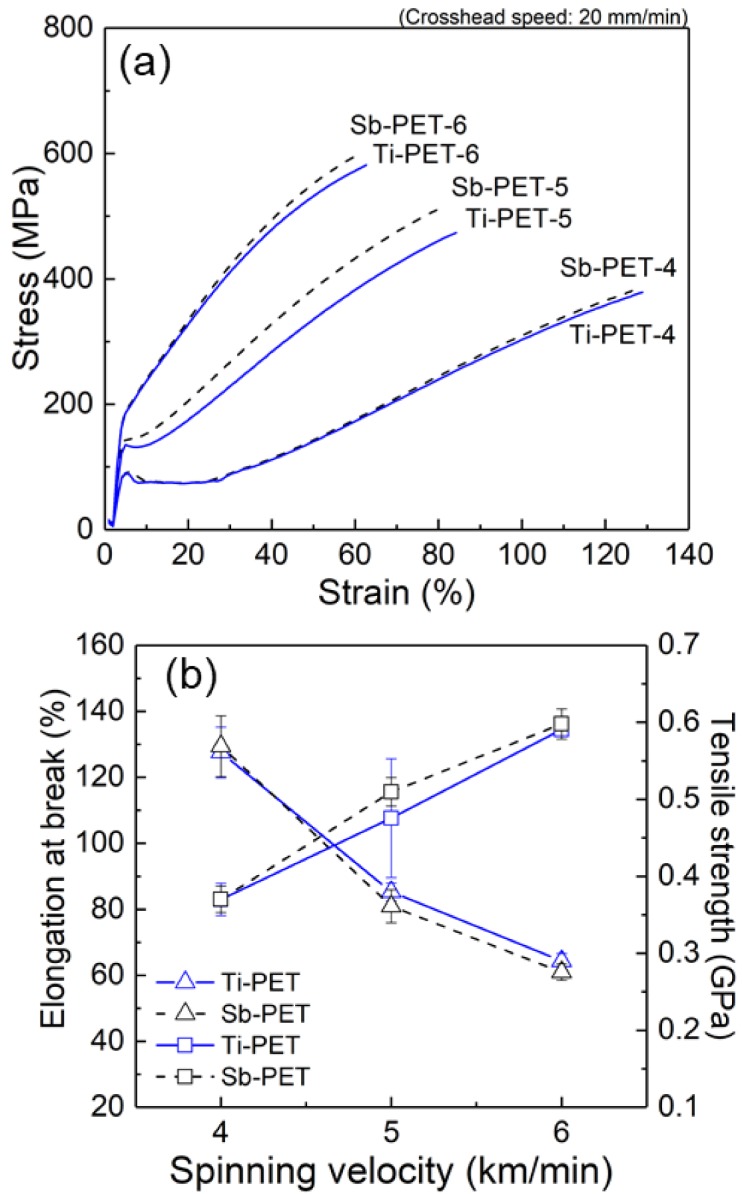
(**a**) Stress–stain curves of the Ti- and Sb-PET as-spun fibers prepared at different spinning velocities; (**b**) Change in elongation at break and tensile strength values of the Ti- and Sb-PET as-spun fibers as a function of spinning velocity (△: Elongation at break, □: Tensile strength).

**Figure 5 polymers-11-01931-f005:**
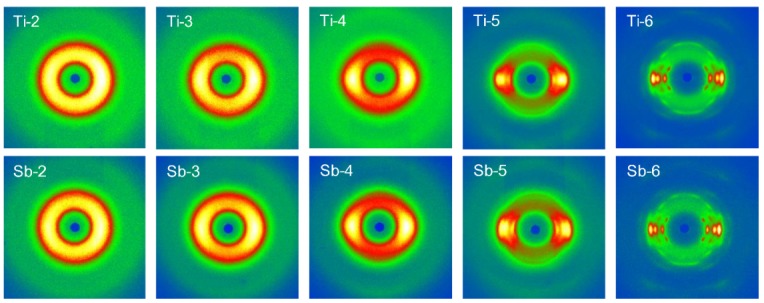
Two-dimensional wide-angle X-ray diffraction (2D-WAXD) patterns of the Ti- and Sb-PET as-spun fibers prepared at different spinning velocities.

**Figure 6 polymers-11-01931-f006:**
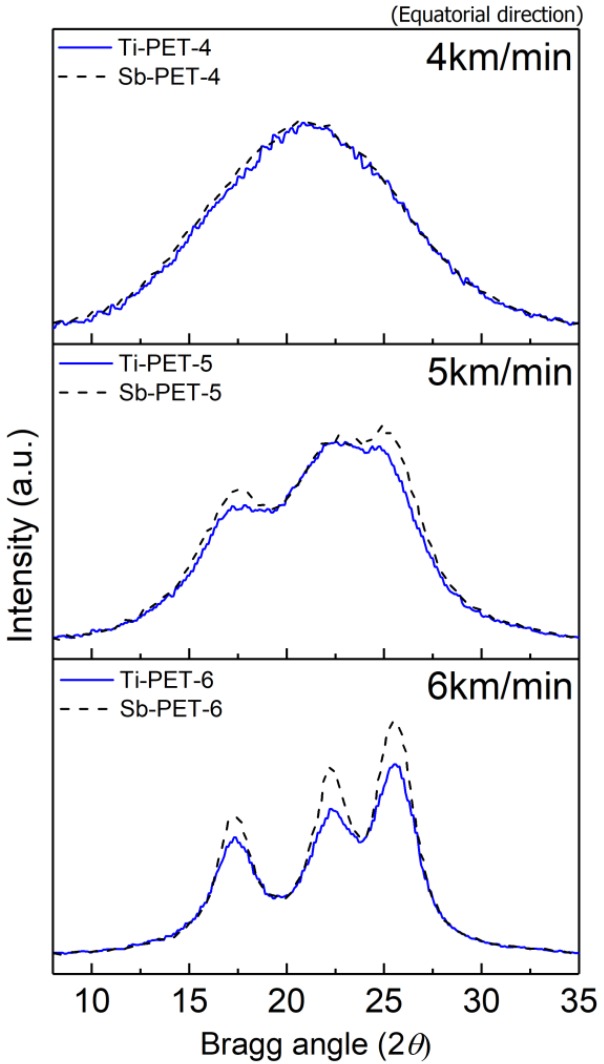
Intensity profiles in the equatorial direction of Ti- and Sb-PET as-spun fibers obtained at various spinning velocities.

**Figure 7 polymers-11-01931-f007:**
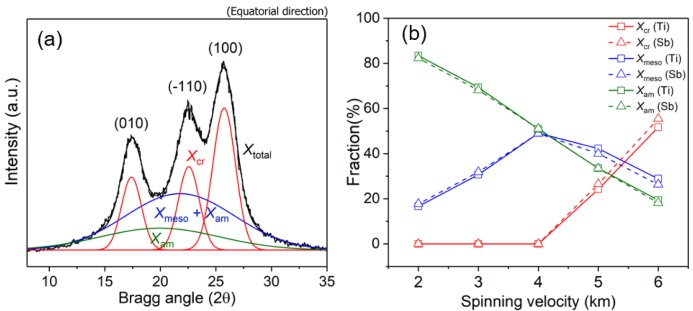
(**a**) Resolution of peaks by fitting the equatorial intensity profile extracted from the 2D-WAXD pattern of the Ti-PET-6 as-spun fiber, where *X_cr_*, *X_meso_*, and *X_am_* indicate the peak areas for the crystalline phase, mesophase, and amorphous phase, respectively; (**b**) change in phase fraction as a function of the spinning velocity.

**Figure 8 polymers-11-01931-f008:**
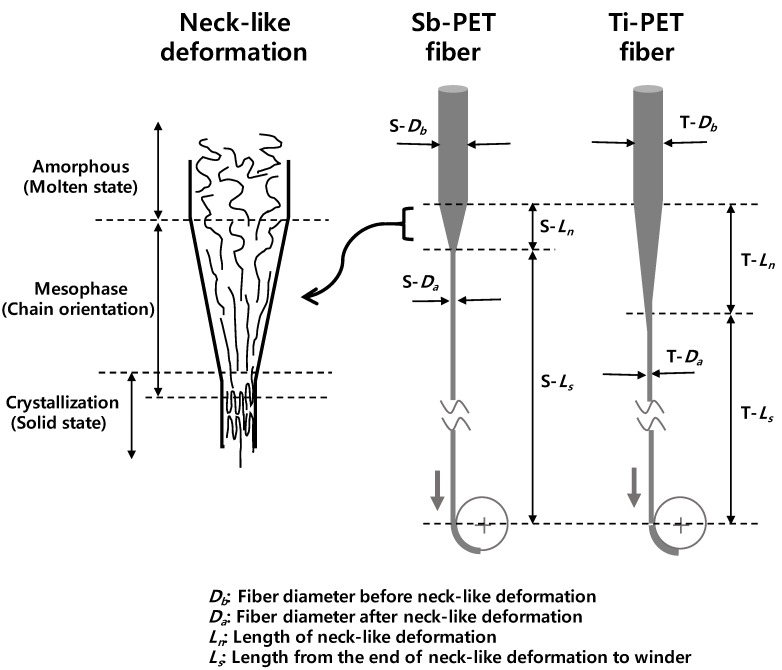
Schematic of neck-like deformation and orientation-induced crystallization in high-speed melt spinning.

**Table 1 polymers-11-01931-t001:** Specimen names of the as-spun Poly (ethylene terephthalate) PET fibers obtained at various spinning velocities.

Spinning Velocity (km/min)	Catalyst	
	TiO_2_	Sb_2_O_3_
2	Ti-PET-2	Sb-PET-2
3	Ti-PET-3	Sb-PET-3
4	Ti-PET-4	Sb-PET-4
5	Ti-PET-5	Sb-PET-5
6	Ti-PET-6	Sb-PET-6

**Table 2 polymers-11-01931-t002:** Characterization results for Titanium based (Ti-) and antimony-based (Sb-) PET resins, and respective as-spun fibers obtained at a spinning velocity of 4 km/min.

Sample		IV (dl/g)	GPC			ICP	TGA ^1^	
			*M* _n_	*M* _w_	*M*_w_/*M*_n_	Sb (ppm)	99%	90%
Ti-PET	Resin	0.634 (0.602 **^2^**)	4153	11,235	2.70	-	369	398
Sb-PET	Resin	0.630 (0.592 **^2^**)	4124	10,986	2.66	244 (163 **^2^**)	372	402

^1^ The temperature at weight loss of 99 wt % and 90 wt %. ^2^ Data of as-spun fiber.

**Table 3 polymers-11-01931-t003:** The crystallization half-time (*t*_½_), Avrami constant (*n*), and crystallization rate (*K*) of Ti- and Sb-PET resins for melt–crystallization at different isothermal temperatures (195 °C, 205 °C).

Sample	Temperature (°C)	*t*_1/2_ (s)	*n*		*K* (s^−1^)	
			*n* _1_	*n* _2_	*K* _1_	*K* _2_
Ti-PET	195	239	1.1	2.7	27.5 × 10^−6^	2.2 × 10^−7^
	205	471	1.4	2.7	4.0 × 10^−6^	0.5 × 10^−7^
Sb-PET	195	139	-	2.4	-	43.0 × 10^−7^
	205	241	1.3	2.5	25.1 × 10^−6^	7.9 × 10^−7^

**Table 4 polymers-11-01931-t004:** Characterization results derived from 2D-WAXD patterns of the Ti- and Sb-PET as-spun fibers prepared at different spinning velocities.

Sample	β_meso_ ^1^	Mass Fraction ^2^ (%)			*d*_spacing_ (Å)			Crystallite Size ^3^ (Å)			No. of Unit Cells ^4^			*f* _c_ ^5^
	(deg)	*X_cr_*	*X_meso_*	*X_am_*	(010)	(100)	(−103)	(010)	(100)	(−103)	(010)	(100)	(−103)	*(010)*
**Ti-PET-2**	39.4	0.0	16.6	83.4	-	-	-	-	-	-	-	-	-	-
**Ti-PET-3**	33.6	0.0	30.7	69.3	-	-	-	-	-	-	-	-	-	-
**Ti-PET-4**	26.7	0.0	49.1	50.9	-	-	-	-	-	-	-	-	-	-
**Ti-PET-5**	24.8	24.2	42.3	33.5	5.22	3.54	3.44	31.3	24.1	39.6	6.0	6.8	11.5	0.893
**Ti-PET-6**	20.1	51.7	28.9	19.4	5.08	3.45	3.40	40.6	34.9	54.7	8.0	10.1	16.1	0.921
**Sb-PET-2**	38.1	0.0	17.7	82.3	-	-	-	-	-	-	-	-	-	-
**Sb-PET-3**	35.2	0.0	31.8	68.2	-	-	-	-	-	-	-	-	-	-
**Sb-PET-4**	26.5	0.0	48.9	51.1	-	-	-	-	-	-	-	-	-	-
**Sb-PET-5**	22.1	26.6	40.0	33.5	5.21	3.51	3.46	30.8	26.1	44.5	5.9	7.4	12.9	0.897
**Sb-PET-6**	18.8	55.4	26.4	18.2	5.09	3.46	3.41	45.7	37.7	55.9	9.0	10.9	16.4	0.925

^1^ Azimuthal peak spread (half-width/2) of oriented mesophase. ^2^ Mass fractions of crystalline phase (*X_cr_*), oriented mesophase (*X_meso_*), and amorphous phase (*X_am_*). ^3^ Analyzed by Scherrer’s method. ^4^ The number of unit cells stacked in the respective plane. ^5^ Crystalline orientation factor.

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
