# Peer review of "Effect of Polycondensation Catalyst on Fiber Structure Development in High-Speed Melt Spinning of Poly (Ethylene Terephthalate)"

_polymers, 2019, doi:10.3390/polym11121931_

Round 1

Reviewer 1 Report

In this paper, authors have synthesized polyethylene terephthalate using a catalyst based on Ti as well as using the more usual, but more toxic, based on Sb. Then, structure and performance of both PET are characterized.

The structure of the paper is adequate and results, their discussion and extracted conclusions are appropriate.

I recommend the publication after a minor revision.

- My principal remark about the content is related with the data given in Table 2. Intrinsic viscosity data suggest that a decrease of the molecular weight occurs during the high-speed melt spinning process. A degradation occurs during this process? However, authors do not give the corresponding data measured by GPC neither TGA. These measurements would clarify this point.

- I have serious problems to distinguish the data corresponding to the different samples given in Figure 2, especially in the case of b) one for the Sb samples. I think that figures b) and c) require the use of four different colors, one for each sample.

- In page 2, line 84, it is the first mention to intrinsic viscosity thus, the complete definition must be written together to its acronym (as authors do in page 3, line 100).

- Authors must take care of the definition of the magnitudes. For instance, they use i) “n” in the Avrami equation as well as for the change of the birefringence and ii) “K” in both Avrami and Scherrer equations. It is true that in this paper the possibility of misunderstandings are low due to different facts: a) in the case of the Avrami equation analysis, since two steps have been observed in the isothermal crystallization, authors give their values corresponding to these magnitudes denoting them with a subscript (ni and Ki); b) in the case of the birefringence authors really use ∆n and c) both definitions of K are discussed in different sections, which are found within the text very far from each other. I do not know if authors have realised that these facts diminishing the possibility of the misunderstanding or it has been by chance. Anyway, authors must take into account the requirement of a precise definition of the magnitudes for further manuscripts in order to avoid these misunderstandings.  

Author Response

Thank you for your kind and extremely helpful comments.

In response to your recommendations, I have revised the manuscript.

Previous studies had produced IV and GPC results with similar tendencies. Therefore, our experiments addressed the IV values only. To avoid confusion, we have completely reworked Table 2. The parts covering the as-spun fiber were removed, with those results being provided as part of remark 2. We initially used four colors for the graph, but this could have led to confusion. To prevent this, we set out to create a group (Sb- and Ti-). We attempted to make the graph easier to understand by increasing the line size and decreasing the data interval. We hope that this makes the graph easier to comprehend. On page 2, we revised line 84 (by inserting “intrinsic viscosity (IV)”). We revised the manuscript in response to each of your comments. In the part addressing birefringence, “∆n” was changed to “∆nB. Furthermore, the “K” in the Scherrer equation was changed to “KS”.

Again, we would like to thank you for your very useful comments.

Reviewer 2 Report

Dear authors,

in my opinin this is an interesting paper of the investigation of the effect of a Ti-based catalyst on the final properties of PET, in comparison to PET obtained using a Sb-based catalyst.

Results are interesting and well presented.

Here below I report some minor changes that, in my opinion, would improve the overall quality of the paper.

Lines 79-82: Please delete this sentence from the Section “Materials” moving it to Introduction or Discussion

Line 102: 5/5 w/w, please check and correct

Abstract, discussion and conclusions.

The sentence reporting that Ti-PET fibers showed a lower tensile strength compared with Sb-PET fibers, under the high-speed spinning conditions, seems not justified considering the experimental results of mechanical analysis (and the standard deviation values). In my opinion, the strength of this sentence must be reduced.

Despite the observed differences underlined by WAXD analysis, Ti-PET fibers show average tensile strength values only slightly lower than those shown by Sb-PET fibers.

I also suggest to move mechanical analysis as this sections discusses results that are less interesting than those reported in the WAXD analysis section.

Author Response

Thank you for your kind and extremely helpful comments.

In response to your recommendations, I have revised the manuscript.

Lines 79–82 were moved to line 50 of the introduction. As a result, the reference order was also revised. Line 102: We checked and corrected the reference to 5/5 w/w. Our original manuscript contained some errors, which we thank you for identifying. The length of the text was reduced and I also changed the expressions.

Again, we would like to thank you for your very useful comments.